


# Applications of matrix factorization methods to climate data

Dylan Harries[1] and Terence J. O'Kane[1]

[1]CSIRO Oceans and Atmosphere, Hobart, Australia

**Correspondence:** Dylan Harries (Dylan.Harries@csiro.au)

**Abstract.** An initial dimension reduction forms an integral part of many analyses in climate science. Different methods yield low-dimensional representations that are based on differing aspects of the data. Depending on the features of the data that are relevant for a given study, certain methods may be more suitable than others, for instance yielding bases that can be more easily identified with physically meaningful modes. To illustrate the distinction between particular methods and identify

circumstances in which a given method might be preferred, in this paper we present a set of case studies comparing the results obtained using the traditional approaches of EOF analysis and $k$-means clustering with the more recently introduced methods such as archetypal analysis and convex coding. For data such as global sea surface temperature anomalies, in which there is a clear, dominant mode of variability, all of the methods considered yield rather similar bases with which to represent the data, while differing in reconstruction accuracy for a given basis size. However, in the absence of such a clear scale separation, as in

the case of daily geopotential height anomalies, the extracted bases differ much more significantly between the methods. We highlight the importance in such cases of carefully considering the relevant features of interest, and of choosing the method that best targets precisely those features so as to obtain more easily interpretable results.

## 1 Introduction

An ubiquitous step in climate analyses is the application of an initial dimension reduction method to obtain a low-dimensional representation of the data under study. This is, in part, driven by the purely practical fact that large, high-dimensional datasets are common, and to make analysis feasible some initial reduction in dimension is required. Often, however, we would like to associate some degree of physical significance to the elements of the reduced basis, for instance by identifying separate modes of variability. Given the wide variety of possible dimension reduction methods to choose from, it is important to understand

the strengths and limitations associated with each for the purposes of a given analysis.

    Perhaps the most familiar example in climate science is provided by empirical orthogonal function (EOF; Lorenz (1956); Hannachi et al. (2007)) or principal component analysis (PCA; Jolliffe (1986)), which identifies directions of maximum variance in the data. The difficulties inherent in interpreting EOF modes physically have been thoroughly documented (Dommenget





and Latif, 2002; Monahan et al., 2009) and partly motivate various modifications of the basic EOF analysis (Kaiser, 1958; Rich-
man, 1986; Jolliffe et al., 2003; Lee et al., 2006; Mairal et al., 2009; Witten et al., 2009; Jenatton et al., 2010).

Another approach to constructing interpretable representations is based on cluster analysis, which, in its simplest variants, identifies regions of phase space that are repeatedly visited (MacQueen, 1967; Ruspini, 1969; Dunn, 1973; Bezdek et al., 1984). The utility of clustering-based methods is founded on the apparent existence of recurrent flow patterns over a range of time-scales (Michelangeli et al., 1995). As estimating the multidimensional probability density function (PDF) associated
with the distribution of states is generally difficult, clustering methods attempt to detect groupings or regions of higher point density, which may in some cases be approximations to peaks in the underlying distribution, and otherwise detect preferred weather patterns or types (Legras et al., 1987; Mo and Ghil, 1988). Extensions to standard clustering algorithms may take into account the fact that such patterns usually exhibit some degree of persistence or quasi-stationarity (Dole and Gordon, 1983; Renwick, 2005). Hierarchical or partitioning based clustering techniques, of which $k$-means is a popular example, have been
widely used to identify spatial patterns associated with regimes or to classify circulation types (see, e.g., Mo and Ghil (1988); Stone (1989); Molteni et al. (1990); Cheng and Wallace (1993); Hannachi and Legras (1995); Kidson (2000); Straus et al. (2007); Fereday et al. (2008); Huth et al. (2008); Pohl and Fauchereau (2012); Neal et al. (2016)). Despite their widespread use, there can be difficulties in interpreting the resulting patterns (Kidson, 1997), while ambiguities in selecting the number of clusters can lead to conflicting characterizations of regimes (Christiansen, 2007). In particular, while $k$-means is widely used
due to its simplicity, if the data does not fall into well-defined, approximately spherical clusters, the method need not provide a particularly useful classification of each sample, in which case an alternative clustering algorithm may be more appropriate.

The output of clustering algorithms such as $k$-means is an assignment of each data point to a cluster, and a collection of cluster centroids corresponding to the mean within each cluster. The result is a partition of the phase space in which the elements of each partition are taken to be well represented by the cluster mean, as in vector quantization applications (Lloyd,
1982; Forgey, 1965). While this can yield a useful decomposition of the data, the ideal case occurring when the data do indeed form well-defined clusters, a representation in terms of cluster means is not always effective or suitable for all applications. For instance, the $k$-means centroids need not be realized as observed points within the dataset[1], and, by virtue of their definition, will generally not afford a good representation of edges or extrema of the observed data. When such features are relevant, a possible alternative is to employ methods that construct a basis from points lying on (or outside of) the convex hull of the data.
An example of this approach is given by archetypal analysis (AA; Cutler and Breiman (1994); Stone and Cutler (1996); Seth and Eugster (2016)), in which the basis vectors are required to be convex linear combinations of the data points that minimize a suitable measure of the reconstruction error. This has the effect of identifying points corresponding to extrema of features within the data. Unlike partitioning based cluster algorithms such as $k$-means, AA does not yield a partition of the data into a set of clusters, so that it is not so straightforward to assign observations to a set of regimes. Instead, each observation is
represented as a linear combination of the reduced basis of archetypes with appropriate convex weights, and is in this sense similar to PCA. Compared to PCA, however, the archetype basis may (in some cases) be more easily interpreted, as the basis

---

[1]A simple example would be a naïve analysis of noisy annular data, for which $k$-means may give rise to centroids lying within the annulus; as noted above, in general the success of all of the methods considered in the following will depend on the underlying topology of the data.





vectors correspond to extreme points of the observed data, or convex combinations thereof, rather than abstract directions in the data space.

Unlike PCA and standard clustering methods, AA has only relatively recently found use in climate studies (Steinschneider and Lall, 2015; Hannachi and Trendafilov, 2017). As a result, there has been little comparison of the results of using AA for dimension reduction versus more traditional approaches. Dimension reduction methods such as PCA, $k$-means, and AA can all be expressed generically as approximately representing the data in terms of some set of basis vectors (Mørup and Hansen, 2012), possibly with an additional stochastic error term[2],

$$\boldsymbol{x}(t) \approx \hat{\boldsymbol{x}}(t) = \sum_{i=1}^{k} z_i(t) \boldsymbol{w}_i + \boldsymbol{\epsilon}(t), \tag{1}$$

where $\boldsymbol{x}(t)$ is the $d$-dimensional observation at time $t$, $\hat{\boldsymbol{x}}(t)$ the corresponding reduced representation, and $\{\boldsymbol{w}_i\}$ is the set of $k$ basis vectors. Each method differs in the definition or criteria used to obtain the basis $\{\boldsymbol{w}_i\}$ and weights $z_i(t)$, and hence the choice of method necessarily plays a key role in the nature of the retained features in the data and the interpretation of subsequent results (Lau et al., 1994). Understanding how the methods differ is important for assessing the appropriateness of each for a given task, and for comparing results between methods. For instance, AA is a natural choice when the salient features are in close correspondence with extremal points, but might be less informative in representing data with a regime-like or clustered structure in which a single cluster must be represented by multiple archetypes. When the data exhibits an elliptical distribution in phase space, modes extracted by PCA are easily interpreted, and $k$-means clustering can also be expected to perform well, but this need not be the case for more complicated distributions. Thus, naïvely we might expect that a clustering-based approach might be more useful for the purpose of identifying recurrent regimes, whereas AA may be a better choice for locating extremes of the dynamics.

Ultimately though, a fuller understanding might be best obtained by using generalizations of the above methods, or combinations of multiple methods. AA can be regarded as a constrained convex encoding (Lee and Seung, 1997) of the dataset, where the basis vectors are restricted to be linear combinations of observed datapoints. Relaxing this constraint allows for a more flexible reduction in which the archetypes may not be representable as convex combinations of the data (Mørup and Hansen, 2012), although the effectiveness of doing so will depend somewhat on the underlying data generating process. The problem of finding a convex coding of a given dataset can be unified with PCA and $k$-means clustering by phrasing each as a generic matrix factorization problem (Singh and Gordon, 2008); we review the basic formulation in Sect. 2. In addition to providing a consistent framework for defining each method, it is straightforward to incorporate additional constraints or penalties, e.g., for the purposes of feature selection or to induce sparsity (Jolliffe et al., 2003; Lee et al., 2006; Mairal et al., 2009; Witten et al., 2009; Jenatton et al., 2010; Gerber et al., 2020). Solving the resulting (usually constrained) optimization problem amounts to learning a dictionary with which to represent the data, with different methods producing different dictionaries. By carefully defining the optimization problem, the learned dictionary can be tuned to target particular features in the data. Below,

---

[2]Probabilistic formulations of PCA and AA have also been proposed (Roweis, 1998; Tipping and Bishop, 1999; Seth and Eugster, 2016), in which the problem is formulated as inference under an appropriate latent variable model; similarly, soft $k$-means is well-known to be closely related to Gaussian mixture modelling, e.g., MacKay (2003). For simplicity, in the following comparisons we will only consider the deterministic formulations of the methods.

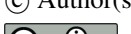



we demonstrate this process by utilizing a recently introduced regularized convex coding (Gerber et al. (2020)), which allows for feature selection to be performed by varying a regularization parameter. By tuning the imposed regularization to optimize

the reconstruction or prediction error, the relative performance of selecting a basis lying on or outside the convex hull can be compared to one that preferentially extracts cluster means.

The purpose of this paper is to explore some of the above issues in the context of climate applications, in the hope that this may provide a useful aid for researchers in constructing their own analyses. In particular, we aim to illustrate some of the strengths and weaknesses of PCA and other dimension reduction methods, using as examples $k$-means, AA, and general

convex coding, by applying each method in a set of case studies. We first apply the methods to an analysis of global sea surface temperature (SST) anomalies, as in Hannachi and Trendafilov (2017). Interannual SST variability is in this case dominated by El Niño-Southern Oscillation (ENSO) activity (Wang et al., 2004), for which there is a large scale separation between this mode and the sub-leading modes of variability, and consequently all methods are effective in detecting this feature of the data. Differences between the methods do arise at smaller scales, however. We then consider a similar analysis of daily 500 hPa

geopotential height anomalies. Unlike SST, the time-series of height anomalies generally does not exhibit noticeably large excursions corresponding to weather extremes; in this case, extremes in the data are characterized not by the amplitude of the anomalies but by their temporal persistence. This demonstrates a limitation of all of the considered methods, namely, that (at least in their basic formulation) persistence or quasi-stationarity is not taken into account. Thus, a direct application of AA or convex codings may not be effective in detecting extremes, while clustering methods may be more informative in detecting

recurrent weather patterns (to the extent that any such regimes are present in the analyzed fields).

The remainder of this paper is structured as follows. In Sect. 2 we review the dimension reduction methods used. In Sect. 3, we compare the results obtained using each method in a set of case studies to illustrate the distinctions between the methods. Finally, in Sect. 4 we summarize our observations and discuss possible future extensions.

## 2 Matrix factorizations

In this section, we first describe the dimension reduction methods that we use in our case studies. As noted above, PCA, $k$-means, and convex coding applied to multidimensional data can all be phrased as matrix factorization problems. Given a collection[3] of $d$-dimensional datapoints $\boldsymbol{x}(t) \in \mathbb{R}^d$, $t = 1, \ldots, T$, we may conveniently arrange the data into a $T \times d$ design matrix $X$ with rows formed by the data samples. In this notation, the reduced representation Eq. (1) becomes

$$X \approx \hat{X} = ZW^T, \tag{2}$$

where $Z \in \mathbb{R}^{T \times k}$ is a $T \times k$ dimensional matrix with rows giving the weights $z_i(t)$, and $W \in \mathbb{R}^{d \times k}$ contains the basis vectors $\boldsymbol{w}_i$ as columns. Note that here and below we assume that the columns of $X$ have zero mean; if not, this can always be arranged

---

[3]In the following, we use notation appropriate for a time-series of observations with separate samples indexed by time $t$, but of course the discussion is not limited to this case.





by first centering the data,

$$X = \left(I_{T\times T} - \frac{1}{T}\mathbf{1}_T\mathbf{1}_T^T\right)\tilde{X},$$

where $\tilde{X}$ denotes the original data, $I_{T\times T}$ the $T\times T$ identity matrix, and $\mathbf{1}_T$ the $T$-dimensional vector of ones.

The factors $W$ and $Z$ are calculated as the minimizers of a suitably chosen cost function $F(W,Z;X)$, measuring (in some application dependent sense) the quality of the reconstruction $\hat{X}$, subject to the constraint that they are within certain feasible regions $\Omega_Z$ and $\Omega_W$,

$$(W,Z) \equiv \operatorname*{arg\,min}_{Z\in\Omega_Z, W\in\Omega_W} F(W,Z;X). \tag{3}$$

A typical cost function takes the form of a decomposable loss function, measuring the reconstruction error, together with a set
of penalty terms imposing any desired regularization, i.e.,

$$F(W,Z;X) = \frac{1}{T}\sum_{t=1}^T \ell(W, \boldsymbol{z}(t); \boldsymbol{x}(t)) + \lambda_W \Phi_W(W) + \lambda_Z \Phi_Z(Z). \tag{4}$$

In Eq. (4) we have, for simplicity, supposed that $W$ and $Z$ are independently regularized, with the tunable parameters $\lambda_W$ and $\lambda_Z$ governing the amount of regularization. Common choices for the loss function include the $\ell_2$-norm[4],

$$\ell(W, \boldsymbol{z}(t); \boldsymbol{x}(t)) \equiv \frac{1}{2}\|\boldsymbol{x}(t) - W\boldsymbol{z}(t)\|_2^2, \tag{5}$$

in which case the unregularized cost is proportional to the residual sum of squared errors (RSS), the Kullback-Leibler divergence for non-negative data,

$$\ell_{KL}(W, \boldsymbol{z}(t); \boldsymbol{x}(t)) = \sum_{i=1}^d \left(x_i(t)\ln\frac{x_i(t)}{(W\boldsymbol{z}(t))_i} - x_i(t) + (W\boldsymbol{z}(t))_i\right), \tag{6}$$

or the wider class of Bregman divergences (Bregman, 1967; Banerjee et al., 2005; Singh and Gordon, 2008). In addition to the soft constraints imposed by the penalty terms $\Phi_W(W)$ and $\Phi_Z(Z)$, the feasible regions $\Omega_Z \subset \mathbb{R}^{T\times k}$ and $\Omega_W \subset \mathbb{R}^{d\times k}$ define
a set of hard constraints that must be obeyed by the optimal solutions. The definition of a given method thus comes down to a small number of modelling choices regarding the cost function and feasible regions.

PCA, in its synthesis formulation, is equivalent to minimizing an $\ell_2$-loss function with $\lambda_W = \lambda_Z = 0$, i.e.,

$$F_{PCA}(W,Z;X) = \frac{1}{2T}\left\|X - ZW^T\right\|_F^2, \tag{7}$$

where $\|A\|_F$ denotes the Frobenius norm of a matrix, subject to the constraint that the reconstruction $\hat{X}$ has rank at most $k$.
The problem in this case has a global minimum (Eckart and Young, 1936) given by the singular value decomposition (SVD), and the retained basis vectors $\boldsymbol{w}_i$ correspond to the directions of maximum variance. In our numerical case studies, we adopt

---

[4]The $\ell_p$-norm of a vector $\boldsymbol{x}\in\mathbb{R}^d$ is given by $\|\boldsymbol{x}\|_p = \left(\sum_{i=1}^d x_i^p\right)^{1/p}$ for $p>0$, while for $p=0$ the $\ell_0$-norm is defined to be the number of non-zero components of $\boldsymbol{x}$.





the convention $Z = U\Sigma/\sqrt{T-1}$, $W = V$ for the PCA factors $W$ and $Z$, where the SVD of $X = U\Sigma V^T$ for real $X$. While the existence of a direct numerical solution for the optimal factorization makes PCA very flexible and easy to apply, the basis vectors may be difficult to interpret, for instance if they have many non-zero components. Sparse variants of PCA that attempt

to improve on this may be arrived at by introducing a sparsity inducing regularization $\Phi_W(W)$, of which a common choice is the $\ell_1$-penalty

$$\Phi_W(W) = \sum_{i=1}^{k} \|\boldsymbol{w}_i\|_1. \tag{8}$$

The partitioning that results from $k$-means clustering may also be written in terms of a factorization $\hat{X} = ZW^T$. Whereas PCA by construction yields an orthonormal basis with which to represent the data, the $k$-means decomposition is in terms of

the cluster centroids and iteratively attempts to minimize the within cluster variance (Hastie et al., 2005). This is equivalent to minimizing an $\ell_2$-cost function, as in Eq. (7), subject to the constraint that the weights matrix $Z$ has binary elements, $Z_{ti} \in \{0,1\}$, and rows with unit $\ell_1$-norm, $\|\boldsymbol{z}(t)\|_1 = 1$. In other words, the optimization is performed within the feasible region

$$\Omega_Z = \{Z \in \mathbb{R}^{T \times k} | Z_{ti} \in \{0,1\}, \quad Z\mathbf{1}_k = \mathbf{1}_T\}. \tag{9}$$

The corresponding basis matrix $W$ then contains the cluster centroids $\boldsymbol{w}_i$ as columns. Although both PCA and $k$-means can be seen to minimize the same objective function[5], the additional constraints in $k$-means clustering mean that finding the exact clustering is NP-hard (Aloise et al., 2009; Mahajan et al., 2012), and heuristic methods must be used instead (e.g., Lloyd (1982); Hartigan and Wong (1979)). These iterative methods are not guaranteed to find a globally optimal clustering, and must be either run multiple times with different initial guesses or combined with more sophisticated global optimization strategies

to reduce the chance of finding only a local minimum.

The convex codings that we employ below, like PCA and $k$-means, are based on minimizing the least squares loss Eq. (7). The least restricted version (Lee and Seung, 1997; Gerber et al., 2020) requires only that the reconstruction lies in the convex hull of the basis, or, in other words, that the weights $Z$ satisfy the constraints

$$\Omega_Z = \{Z \in \mathbb{R}^{T \times k} | Z \succeq 0, \quad Z\mathbf{1}_k = \mathbf{1}_T\}. \tag{10}$$

where the condition $A \succeq 0$ indicates an elementwise inequality. In the absence of any hard constraints on the basis $W$, the optimization problem to be solved reads

$$(W, Z) = \underset{Z \in \Omega_Z}{\arg\min} \left[ \frac{1}{2T} \left\| X - ZW^T \right\|_F^2 + \lambda_W \Phi_W(W) \right], \tag{11}$$

with $\Omega_Z$ as in Eq. (10); note that, if these constraints are further tightened to require that the columns of $Z$ be non-negative and orthonormal, $Z^T Z = I_{k \times k}$, solving this optimization problem (in general, for $\lambda_W = 0$) would yield the hard $k$-means

clustering of the data (Li and Ding, 2006). In the absence of any regularization ($\lambda_W = 0$), it follows from the Eckart-Young

---

[5]See Ding and He (2004) for additional discussion of the relationship between PCA and $k$-means.





theorem that for a given basis size $k$ the reconstruction error achieved by PCA is always no larger than that achieved by this convex coding, which in turn achieves a residual error that is smaller than that for $k$-means. Thus, were the goal to simply achieve the optimal least squares reconstruction error for a given basis size, PCA remains the preferred choice. The more constrained decomposition implied by performing a convex coding may yield advantages in terms of interpretability or feature

selection. The choice of regularization $\Phi_W$ provides further flexibility in this respect, for instance, an $\ell_1$-penalty as in Eq. (8) can be used to induce sparsity, while Gerber et al. (2020) suggest a penalty term of the form

$$\Phi_W(W) = \frac{1}{dk(k-1)} \sum_{i,j=1}^{k} \|\boldsymbol{w}_i - \boldsymbol{w}_j\|_2^2, \tag{12}$$

observing that this places an upper bound on the sensitivity of the reconstruction to changes in the data. It is worth noting that, when combined with an $\ell_2$-loss function, as the regularization $\lambda_W \to \infty$ all of the basis vectors $\boldsymbol{w}_i$ reduce to the global mean

of the dataset.

For $\lambda_W = 0$, the optimal basis vectors $\boldsymbol{w}_i$ produced by solving Eq. (11) will tend to lie on or outside of the convex hull of the data. Standard AA adds a further more conservative constraint to force the $\boldsymbol{w}_i$ to only lie on the convex hull, thus requiring that the archetypes are realizable in terms of convex combinations of the observed data. Under the more relaxed constraints, while the overall residual error might be reduced, one may extract basis vectors that, in reality, never occur and so are difficult

to make sense of. In AA, the additional constraint amounts to requiring that $W = X^T C^T$ for some non-negative $C \in \mathbb{R}^{k \times T}$ with unit $\ell_1$-norm rows, i.e., $C \in \Omega_C$ with

$$\Omega_C = \{C \in \mathbb{R}^{k \times T} | C \succeq 0, C\mathbf{1}_T = \mathbf{1}_k\}. \tag{13}$$

The corresponding optimization problem reads

$$(C,Z) = \underset{C \in \Omega_C, Z \in \Omega_Z}{\arg\min} \frac{1}{2T} \|X - ZCX\|_F^2, \tag{14}$$

with $\Omega_Z$ as in Eq. (10). The impact of these different choices of cost function and constraints are sketched in Fig. 1.

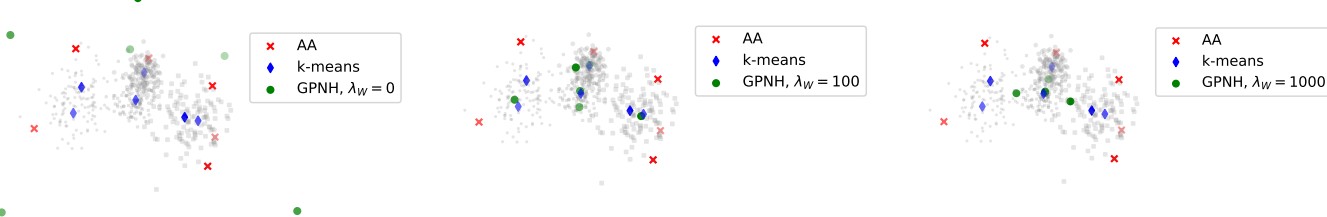

**Figure 1.** Illustration of the different decompositions obtained using $k$-means, regularized convex coding (denoted GPNH), and AA, and of the impact of the regularization parameter $\lambda_W$ when using the penalty term Eq. (12). For increasing (from left to right) $\lambda_W = 0, 100, 1000$, the number of selected features progressively decreases and the basis varies from lying outside the convex hull of the data to the global mean.


The optimization problem posed in either the convex coding case or by AA cannot be solved analytically, as in $k$-means clustering. Moreover, the combined cost function is not convex in the full set of variables $W$ and $Z$ (or $C$ and $Z$ for AA). It is,





though, convex in either $W$ or $Z$ separately, when the other is held fixed, and local stationary points may be straightforwardly found by alternating updates of the basis and weights. For instance, using the penalty function Eq. (12), we may update $W$

with fixed $Z$ via

$$W \leftarrow X^T Z \left[ Z^T Z + \frac{4T\lambda_W}{dk(k-1)} \left( kI_{k \times k} - 1_{k \times k} \right) \right]^{-1}. \tag{15}$$

For fixed $W$, $Z$ may then be updated by projected gradient descent. An alternative that avoids having to perform direct projection onto the simplex (Mørup and Hansen, 2012) is to reparameterize the latent weights $Z$ as

$$Z_{ti} = \frac{H_{ti}}{\sum_{i=1}^{k} H_{ti}}, \quad H_{ti} \geq 0, \tag{16}$$

which automatically satisfies the stochastic constraint on $\boldsymbol{z}(t)$, and $H_{ti}$ is required only to be non-negative, making the necessary projection trivial. The factors $W$ and $H$ may then be updated via, e.g., a sequence of projected gradient descent update steps of the form

$$\nabla_W F(W, Z; X) \leftarrow -\frac{1}{T}(X^T Z) + \frac{1}{T} W \left[ Z^T Z + \lambda_W G_W \right], \tag{17a}$$

$$W \leftarrow W - \eta_W \nabla_W F(W, Z; X), \tag{17b}$$

$$\nabla_Z F(W, Z; X) \leftarrow -\frac{1}{T} XW + \frac{1}{T} ZW^T W, \tag{17c}$$

$$H_{ti} \leftarrow \max \left\{ Z_{ti} - \eta_H \left[ (\nabla_Z F(W, Z; X))_{ti} - \sum_{j=1}^{k} Z_{tj} (\nabla_Z F(W, Z; X))_{tj} \right], 0 \right\}, \tag{17d}$$

$$Z_{ti} \leftarrow \frac{H_{ti}}{\sum_{i=1}^{k} H_{ti}}, \tag{17e}$$

where, for simplicity, we have assumed that the derivative of the penalty term $\Phi_W$ may be written in the form

$$\frac{\partial \Phi_W}{\partial W} = \frac{1}{T} W G_W,$$

which is the case if, for instance, $\Phi_W \propto \mathrm{Tr} \left[ W G_W W^T \right]$ for some symmetric matrix $G_W$. The step-size parameters $\eta_W$ and $\eta_H$ may either be fixed or determined by performing a line-search. This procedure may be iterated until the successive changes in the total cost fall below a given tolerance. As convergence to a global minimum is not guaranteed, in practice this procedure is repeated for multiple initial guesses for $W$ and $Z$ to try to improve the likelihood of locating the optimal solution. Inspecting the update equations Eq. (17), the cost per iteration is seen to scale as $O(dTk) + O(k^2T) + O(k^2d) + O(kd) + O(kT)$. In

particular, in the usual case of interest with $d, T \gg k$, the leading contribution to the cost is linear in each of $d$, $T$, and $k$, i.e., $O(dTk)$, making the method suitable for large datasets and comparable with $k$-means and similar decompositions (Mørup and Hansen, 2012; Gerber et al., 2020).

## 3 Case studies

Having specified the dimension reduction methods, we now turn to a set of case-studies to explore the implications of the

different choices made in their definitions.





### 3.1 Sea surface temperature data

Following Hannachi and Trendafilov (2017), we first apply the methods described in Sect. 2 to monthly SST data. The source of the data is the Hadley Centre Sea Surface Temperature dataset (HadISST), version 1.1 (Rayner et al., 2003), consisting of monthly SST values on a $1° \times 1°$ global grid spanning the time period from January 1870 to December 2018. Monthly

anomalies are calculated by removing from the full time-series a linear warming trend and additive seasonal component, where the annual cycle is estimated based on the 1981 to 2010 base period. The analysis region is restricted to the region between $45.5°$N and $45.5°$S.

As the standard and most familiar method, we first perform PCA on the SST anomalies over the time period January 1870 to December 2018 to establish a baseline set of modes. The anomalies at each grid cell are area weighted by the square root

of the cosine of the point's latitude. To provide a rough measure of the out-of-sample reconstruction error[6], the EOFs and PCs are evaluated on the first 90% of the anomaly time-series (i.e, January 1870 to February 2004), and the root-mean-square error (RMSE) defined by

$$\text{RMSE}_{\text{train/test}}(k) = \sqrt{\frac{1}{dT} \sum_{i=1}^{d} \sum_{t=1}^{T} [x_i(t) - \hat{x}_i(t)]^2} \qquad (18)$$

is computed for the training set and for the test set consisting of the remaining 10% of the data. In Eq. (18), $\hat{x}_i(t)$ is the $i^{\text{th}}$

dimension of the reconstruction from $k$ EOFs, and $T$ refers to the size of the training or test set, as appropriate. We consider the results of retaining the first $k$ modes for $k = 1, \ldots, 40$; for reference, the first 40 modes account for approximately 85% of the total variance. The fraction of the total variance in the training set associated with the leading 10 modes is shown in Fig. 2. The most obvious feature of this variance spectrum is the well-known separation between the first and subsequent modes, with the first mode associated with ENSO variability on interannual time-scales and large spatial scales. This is evident in the spatial

patterns of the EOFs shown in Fig. 3, in which the first mode shows the canonical ENSO pattern of SST anomalies, while the higher order modes correspond to spatially smaller scale variability.

With the above EOF patterns as a point of reference, we now turn to comparing the representation of the dataset produced by each of $k$-means, archetypal analysis, and convex coding. In each case, the same dataset (i.e., latitude weighted, detrended anomalies) is used as input to the dimension reduction method. The individual data points consist of anomaly maps at each

time-step, in other words, the dimension reduction is performed in the state space rather than the sample space (Efimov et al., 1995), such that each dictionary vector corresponds to a particular spatial pattern of SST anomalies. We fit each method using dictionaries of size $k = 1, \ldots, 20$ on the first 90% of the dataset, using the last 10% of samples to provide a simple picture of out-of-sample performance.

---

[6]Of course, in practice proper estimates of the out-of-sample performance would be obtained by an appropriate cross-validation procedure or similar. However, as here we are primarily interested in the qualitative differences between the different methods in terms of extracting recognizable states, we do not focus on the technical details of model selection or optimizing predictive performance, and simply present these out-of-sample estimates to show the general features of each method.

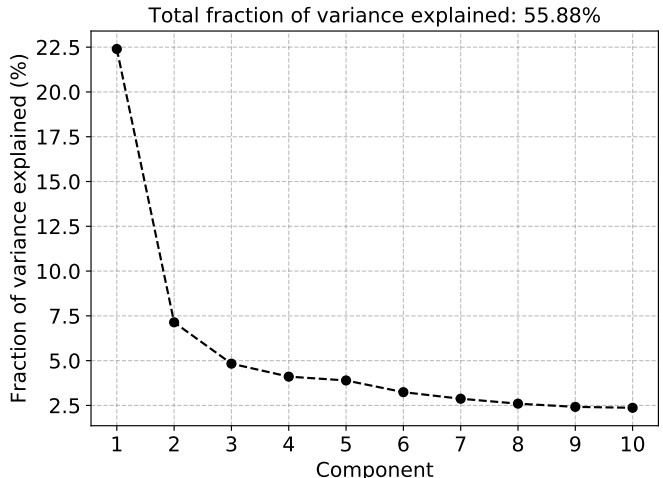

**Figure 2.** Fraction of variance explained by the first 10 modes obtained from PCA of SST anomalies.

We consider the results of a $k$-means clustering analysis of the data first. For each value of $k$, the algorithm is restarted 100 times with different initial conditions and the partitioning found to have the lowest reconstruction error is chosen. As is typical of clustering procedures where the number of clusters must be specified a priori, determining the most appropriate choice of $k$ is non-trivial. One commonly used heuristic is to inspect a "scree" plot of the within-cluster sum of squares (Tibshirani et al., 2001),

$$W_k = \sum_{i=1}^{k} \frac{1}{|C_i|} \sum_{t_1,t_2 \in C_i} \|\boldsymbol{x}(t_1) - \boldsymbol{x}(t_2)\|_2^2, \tag{19}$$

where $|C_i|$ is the size of cluster $C_i$, as a function of the number of clusters. The preferred number of clusters $k^*$ is identified as the location of an elbow or kink in this curve, if any such feature is present. Alternatively, various indices (see, e.g., Arbelaitz et al. (2013) for a review) or Monte Carlo procedures, such as the gap statistic (Tibshirani et al., 2001), have been proposed for assessing whether a given $k$ is suitable. For the present application, plots of the normalized within-cluster sum of squares and the gap statistic computed using 100 Monte Carlo experiments for each $k$ with a null model generated by PCA are shown in Fig. 4. The plot of $W_k$ as a function of $k$ does not show an obvious elbow at any particular $k \leq 20$; similarly, using the gap statistic curve one would conclude that $k = 1$ cluster is preferred[7]. This simply reflects the fact that the anomaly data does not form well-separated clusters (with respect to the assumed Euclidean distance) in the full state-space, making interpreting the different clusters as dynamically distinct regimes difficult, although this does not preclude using the clustering model as a discrete representation of the full dynamics (e.g., Kaiser et al. (2014)).

The partitioning provided by a simple clustering method such as $k$-means may also still be useful in classifying samples so long as proximity in state space, or similarity more generally, carries meaningful information for an application. In the

---

[7]We note that, when using a null model generated by PCA, the gap curve in Fig. 4 might be considered to indicate possible clustering into 5 clusters within the single, large cluster. However, this conclusion depends strongly on the precise null model used to generate the reference data.


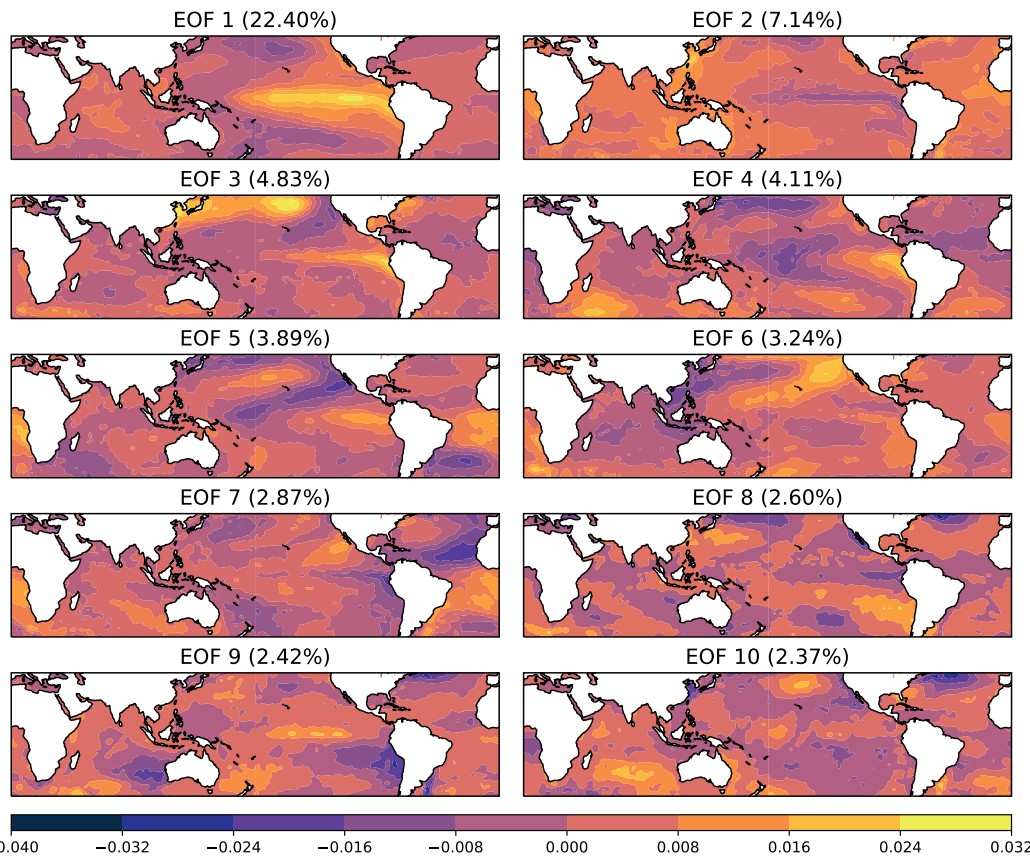

**Figure 3.** Spatial patterns for the leading 10 modes obtained from PCA of SST anomalies.

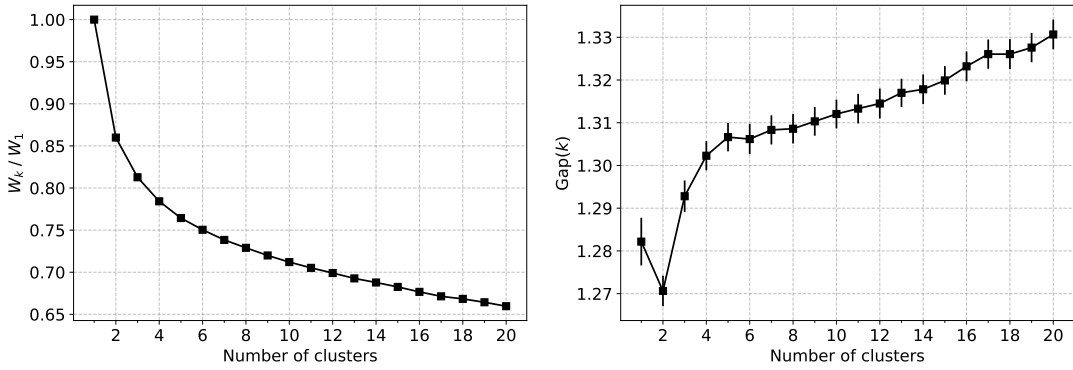

**Figure 4.** Plots of the normalized within-cluster sum of squares (left), $W_k$, and the corresponding gap statistic $\mathrm{Gap}(k)$ (right) for $k$-means clustering of global SST anomalies.


case of detrended SST anomalies, the magnitudes and spatial distribution of the anomalies are of themselves informative, and key drivers such as ENSO manifest as relatively large excursions from anomalies associated with higher-frequency variability. Consequently, a $k$-means clustering of the anomalies does result in a partition in which the cluster centroids, or a subset thereof,

can be identified with physical modes, as the algorithm finds several clusters consisting of these extremes of the point cloud while the remainder partition the bulk of the data. The simplest, if somewhat trivial, case for which this can be seen is for $k = 3$ clusters, shown in Fig. 5. Two of the plotted centroids are recognizable as canonical El Niño and La Niña patterns, while

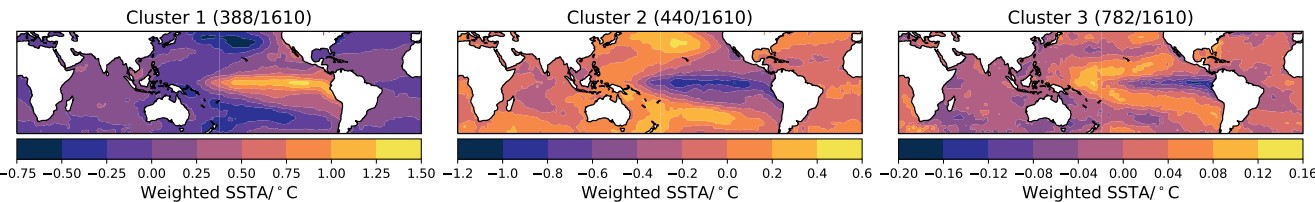

**Figure 5.** Spatial patterns of the cluster centroids obtained from a $k$-means clustering of SST anomalies with $k = 3$. The number of months assigned to each cluster is shown above each map.

the last corresponds to a near-climatological state. The former two clusters are dominated by months at the ends of the first principal axis (i.e., along the leading EOF). The fact that two clusters are required to represent both phases of the leading EOF

is due to the fact that the overall sign of the cluster centroids, which correspond to points in the data space, is meaningful, unlike in PCA; note that the same is true for the convex coding methods to be considered below. To obtain some sense of the relative distributions of the points within each of the clusters, in Fig. 6 we show the projection of the data into two-dimensions generated by metric multidimensional scaling (MDS). The points assigned to the first and second cluster are those furthest from

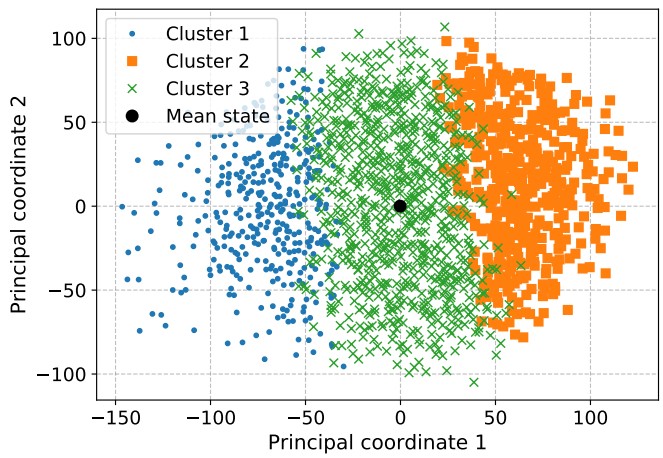

**Figure 6.** Two-dimensional projection of HadISST SST anomalies obtained by metric MDS with a Euclidean distance measure. The assignment of each point to the clusters produced by $k$-means clustering with $k = 3$ in Fig. 5 is also shown.





the mean state, while the remaining cluster accounts for the bulk region. The difference between the first and second centroids
is closely aligned with the leading EOF. Similar behavior results when a larger number of clusters is specified.

The fact that, due to the dominating role of ENSO variability, a $k$-means clustering results in several clusters corresponding
to large magnitude anomalies in turn implies that the corresponding centroids will be relatively close to the convex hull of
the anomaly point cloud. Consequently, these centroids will also closely resemble a subset of the archetypes derived from
an archetypal analysis of the same data, at least qualitatively. Relaxing the requirement that the dictionary vectors lie on the
convex hull, one may still expect that a convex coding applied to the data will identify features along the same directions, albeit
with inflated magnitude so as to reduce the resulting reconstruction error. To verify this, we perform a standard archetypal
analysis and a regularized convex coding of the detrended anomalies, in each case restarting the optimization algorithm 100
times and choosing the best encoding from the multiple starts. The regularization in Eq. (12) is used; to illustrate the effect
of this regularization, we show results for $\lambda_W = 0$ (i.e., no regularization) and $\lambda_W = 10^3$. The fitted archetypes for $k = 3$ are
shown in Fig. 7, and the corresponding dictionary vectors for the regularized convex coding are shown in Fig. 8. The patterns

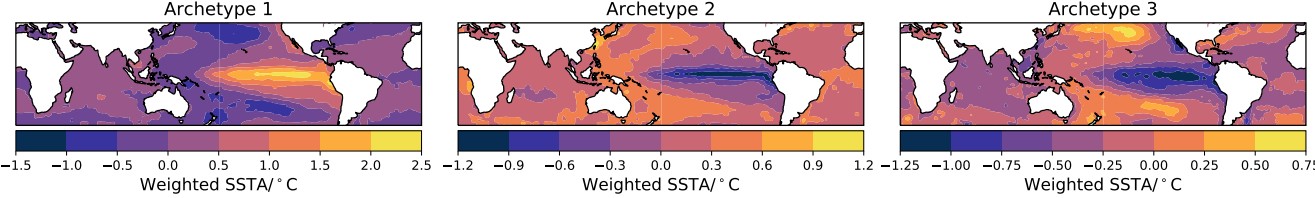

**Figure 7.** Spatial patterns of the archetypes found from AA with $k = 3$ archetypes.

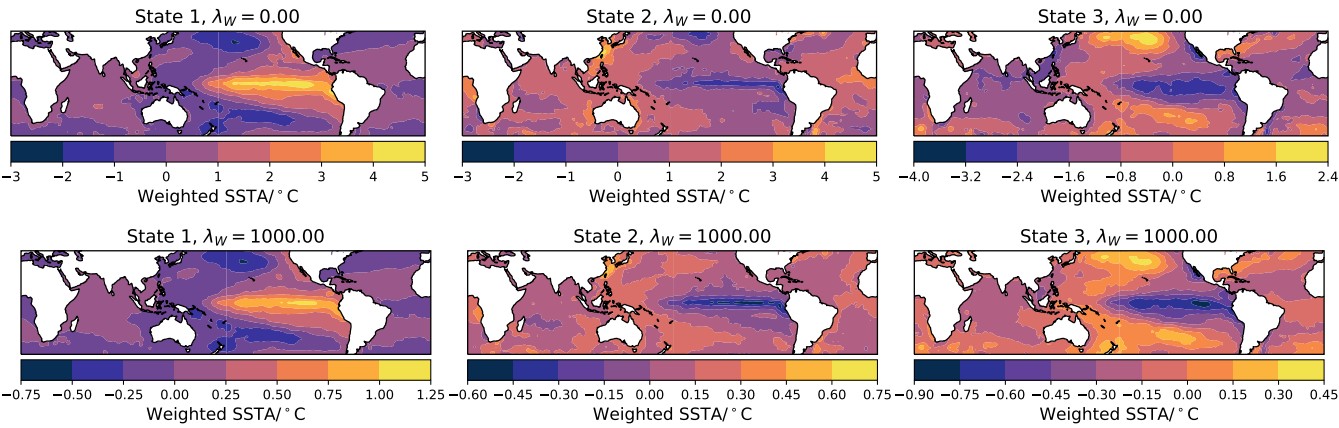

**Figure 8.** Spatial patterns for the convex coding dictionary vectors for $\lambda_W = 0$ (top) and $\lambda_W = 10^3$.

obtained using the two methods in Fig. 7 and Fig. 8 are very similar; in both cases, El Niño and La Niña patterns are found
together with a third configuration containing a cold tongue in the equatorial Pacific with positive anomalies to the north and
south, as noted by Hannachi and Trendafilov (2017). However, the magnitude of the anomalies is substantially larger when



using an unregularized convex coding as the fitted dictionary vectors are chosen to sit well outside the observed point cloud.

While the states correspond to very similar relative signs of anomalies, it is arguably more difficult to interpret the individual dictionary vectors found by the unregularized convex coding physically, as the large anomalies represent far more extreme states than are observed in the data. On the other hand, this also permits a much smaller reconstruction error for a given basis size, and so may be preferable when a higher fidelity reconstruction is required[8]. The effect of the regularization Eq. (12) is to penalize over-dispersion of the dictionary vectors, and so select features that are insensitive to small variations in the input data.

For sufficiently large $\lambda_W$, the resulting states lie on or within the convex hull of the data, and as a result are more comparable to the states found by AA or $k$-means. This is illustrated in Fig. 9, from which it can be seen that the basis vectors produced by the regularized convex coding are closer (in Euclidean distance) to those produced by AA than the unregularized basis vectors.

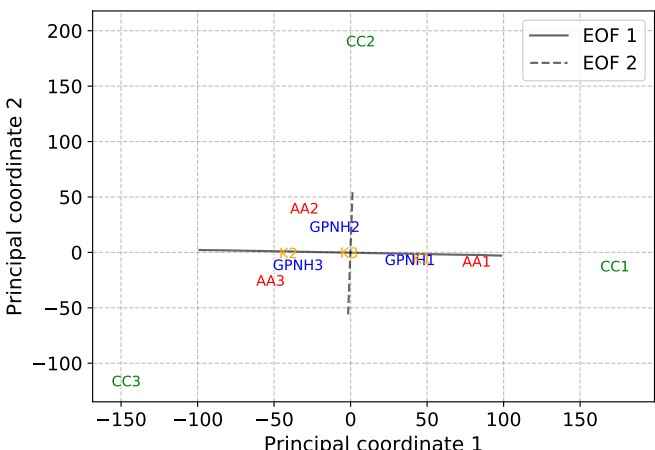

**Figure 9.** Two-dimensional projection of basis vectors obtained by AA, convex coding, and $k$-means based on a metric MDS analysis with Euclidean dissimilarities. The projected locations of the basis vectors for each method are indicated by the labels AA, CC, GPNH, and K for AA, convex coding with $\lambda_W = 0$, convex coding with $\lambda_W = 1000$, and $k$-means, respectively. For reference, the images under the MDS projection of line segments lying along the directions of the first and second EOFs, with lengths proportional to the variance explained by each, are also shown.

        This trade-off between reproducing the data with small errors and constraining the basis vectors to be close to the observed

data is also evident in Fig. 10, where the reconstruction RMSE within the training and held-out test datasets for each of the methods are plotted as a function of the dimension of the reduced order representation. In the absence of any regularization, the RMSE produced by a convex coding of the data is close to the globally optimal result obtained from PCA. For $\lambda_W = 10^3$, the obtained RMSE is larger and similar to that for AA, while $k$-means leads to the largest errors due to the very coarse representation of each datapoint by the centroid of its assigned cluster. It follows that, for a given number of basis vectors,

---

[8]Similar remarks can be made for the AA/PCH-$\delta$ model proposed in Mørup and Hansen (2012).





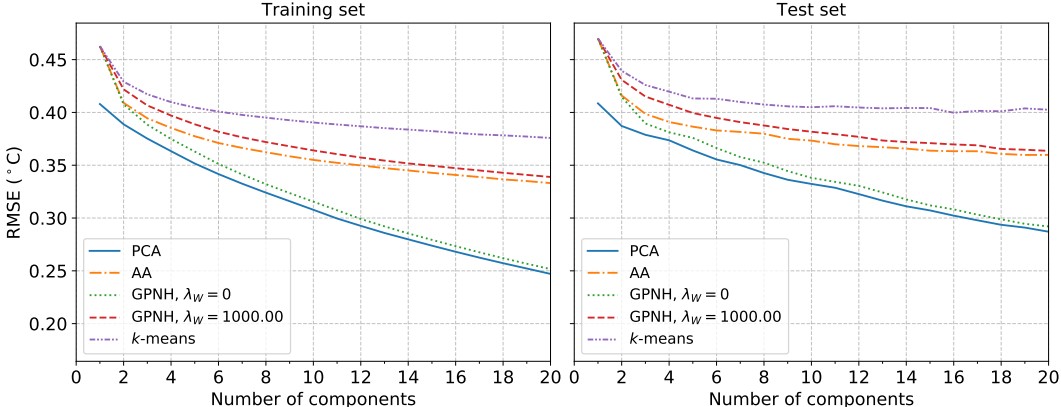

**Figure 10.** Training and test set RMSE for the reconstruction of SST anomalies resulting from each of the methods.

an unregularized convex coding yields performance approaching that of PCA in terms of reconstruction errors, while AA and $k$-means in turn are more constrained and hence reproduce the data with somewhat larger errors.

## 3.2 Geopotential height anomalies

SST anomalies are an example of a dataset in which a dominant mode is well separated in scale from subleading modes of variability. As a result, all of the dimension reduction methods that we consider extract similar bases with which to represent the data, and these can be identified with well-known physical modes. Moreover, physically interesting events such as extremes correspond to large magnitude anomalies relative to the mean state, i.e., at the boundary of the point cloud, and so can be directly extracted by those methods that look for dictionary vectors in the convex hull of the data. This is not true for many variables of interest, however, and so we now compare the behavior of the methods when applied to data that do not exhibit these features.

We consider Northern Hemisphere (NH) daily mean anomalies of 500 hPa geopotential height, $Z'_{g500\text{hPa}}$, between 1 January 1958 and 31 December 2018. The geopotential height data used are obtained from the Japanese 55-year Reanalysis (JRA-55, Kobayashi et al. (2015); Harada et al. (2016)). Anomalies are formed by subtracting the climatological daily mean based on the 1 January 1981 to 31 December 2010 reference period. Unlike monthly SST, there is no clear scale separation in the resulting time series; in Fig. 11 we show the leading EOFs and the percentage of the total variance explained by each. Additionally, the characterization of regimes is more complicated. In particular, physically significant features in the height field are expected to be quasi-stationary or persistent (Michelangeli et al., 1995; Dole and Gordon, 1983; Mo and Ghil, 1988), and are not solely distinguished by their location in the state space. Such features may therefore be better identified as modes of the state space PDF, arising either due to longer residency times or frequent recurrence of particular states. Where this gives rise to regions of higher density in state space, clustering algorithms such as $k$-means might be expected to reasonably well represent the corresponding regimes. Convex reduction methods, on the other hand, default to being less sensitive to recurrence; as in the

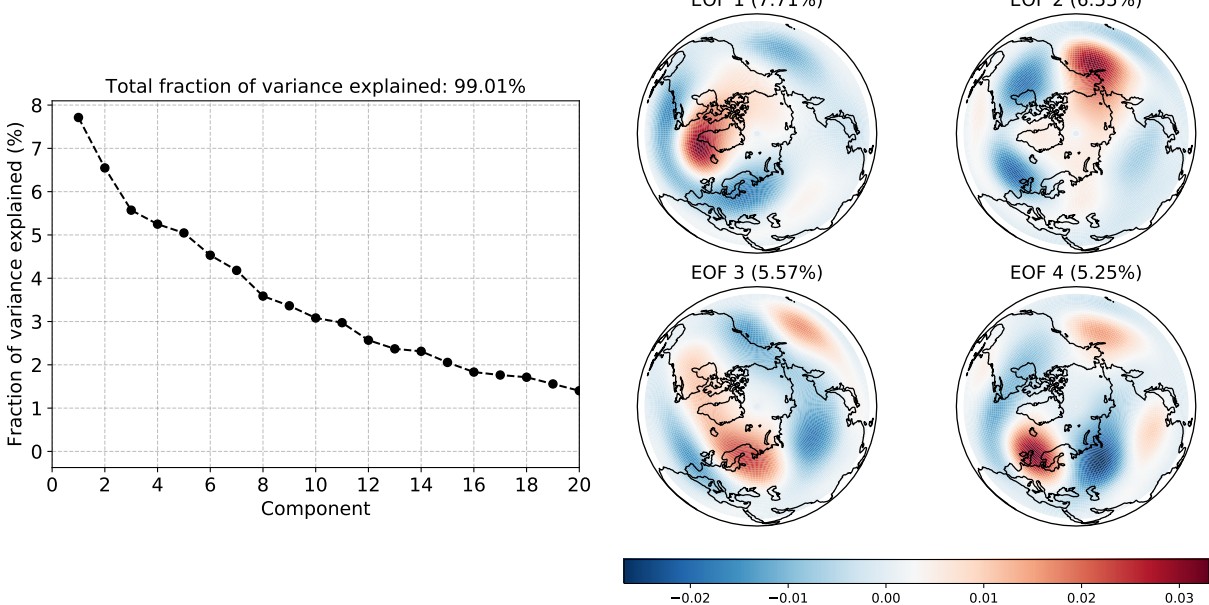

**Figure 11.** Fraction of the total variance associated with each of the first 20 of 167 retained EOF modes of daily NH 500 hPa geopotential anomalies (left), and the spatial patterns associated with the leading 4 modes. The 167 retained modes account for approximately 99% of the variance.

preceding SST example, the fitted dictionary vectors will correspond to points near the boundary of observed data. In doing so, AA and methods based on convex coding preferentially represent the data in terms of deep, but potentially infrequent or highly transient, lows or highs. While this remains an adequate representation simply for reconstructing the full observations, the resulting basis may be more difficult to relate to traditionally identified metastable atmospheric regimes, for instance.

In the case of NH geopotential height anomalies, the cluster centroids, archetypes, and dictionary basis vectors that result from applying $k$-means, AA, and convex coding to the leading 167 PCs[9] of $Z'_{g500\text{hPa}}$ are shown for $k = 4$ clusters in Fig. 12. Note that inspection of the scree plots (not shown) does not indicate a strong preference for a given number of clusters or states for $k \leq 20$, and we choose $k = 4$ as a simple example. The relative dispositions of each of the states with respect to each other, as measured by Euclidean distance, are visualized in Fig. 13 on the basis of a two-dimensional metric MDS. Unlike the SST

case, the $Z'_{g500\text{hPa}}$ data do not exhibit a dominant axis of variability, i.e., there is no clear scale separation between modes. In Fig. 13 this manifests as the absence of a clearly preferred axis along which the basis vectors are distributed[10], c.f., Fig. 9. As the variance in the data is not dominated by a single principal axis, the basis vectors extracted by each of the methods are more evenly distributed around the point cloud, in turn leading to greater differences between the fitted bases.

---

[9]Clustering on the PCs was done so as to reduce the overall cost of the methods; we have checked that, for small numbers of clusters, the spatial patterns that result are very similar.

[10]Similar behavior is evident for different numbers of states, e.g., for $k = 3$ the MDS projection results in a triangular arrangement of points with the climatological point located close to the centroid of the resulting shape.


**Figure 12.** Spatial patterns of geopotential height anomalies corresponding to the $k$-means centroids (first column), archetypes (second column), convex coding basis vectors with $\lambda_W = 0$ (third column), and convex coding basis vectors with $\lambda_W = 10$.

While all methods identify a feature that is strongly reminiscent of the NAO, there is somewhat more variation in the
remaining representative states, in contrast to the case of SST anomalies. In particular, the centroids identified by $k$-means
are no longer in close correspondence with the representative states constructed by either AA or an unregularized convex


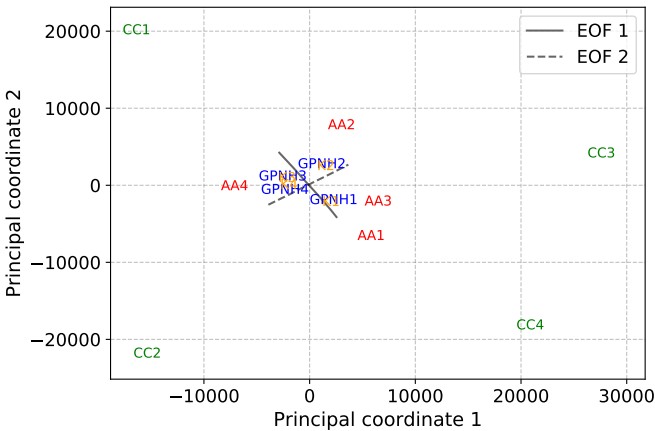

**Figure 13.** Two dimensional projection of spatial patterns of geopotential height anomalies obtained using the various dimension reduction methods by metric MDS. The results of transforming line segments lying along the directions of the first and second EOFs, as in Fig. 9, are also shown.

coding. The $k$-means centroids are characterized by relatively small magnitude anomalies with positive amplitude at longitudes associated with blocking (Pelly and Hoskins, 2003); in particular, the location of the anticyclonic anomaly in cluster 3 in Fig. 12 closely coincides with the center of action of the EU1 (Barnston and Livezey, 1987) or SCA (Bueh and Nakamura, 2007) pattern associated with Scandinavian blocking. The archetypes, in comparison, appear to exhibit a more pronounced wave-train structure, in addition to corresponding to generally larger magnitude anomalies. As expected, the basis obtained via an unregularized convex coding represents far larger anomalies again, defining representative states that are much more extreme than the bulk of the observed daily anomaly fields. The role of the regularization in feature selection is clearly illustrated by comparing the two right-most columns of Fig. 12. By increasing the regularization parameter, the method can be tuned to place less emphasis on capturing large amplitude, noisy variations and target less sensitive features in the data. In this case, for $\lambda_W \approx 1$ the convex coding basis vectors are in close agreement with the archetypes, while for $\lambda_W \approx 10$ they essentially coincide with the $k$-means centroids. In this sense, by appropriate choice of regularization it is possible to interpolate between a representation of the data in terms of points on the convex hull and mean features, depending on how much weight is placed on minimizing the reconstruction error.

In the absence of any regularization, the patterns obtained by a convex coding and by AA correspond to extreme departures from the mean state, but cannot necessarily be directly interpreted as individually representing particular physical extremes. As noted above, this arises due to the fact that such extremes are not necessarily associated with boundaries in state space but may instead be due to extended residence or persistence of a given (non-extreme) state. This difficulty in directly relating atmospheric extremes to a basis produced by AA or similar methods can be clearly demonstrated by considering the representation of a given event in terms of this basis. A dramatic example is provided by the 2010 summer heatwave in western Russia that saw an extended period of well above-average daily temperatures and poor air quality, and was associated with substantial


excess mortality and economic losses (Barriopedro et al., 2011; Shaposhnikov et al., 2014). The upper-level circulation during July 2010 was characterized by persistent blocking over eastern Europe (Dole et al., 2011; Matsueda, 2011). The associated monthly mean pattern of height anomalies for that month (see, e.g., Figure 2 of Dole et al. (2011)) most closely resembles

the pattern for cluster 3 obtained with $k$-means in Fig. 12. Consistent with this is the fact that most days during July 2010 are assigned to this cluster, as shown in Fig. 14. Consequently, cluster 3 might be reasonably well interpreted as representing (a class of) European blocking events.

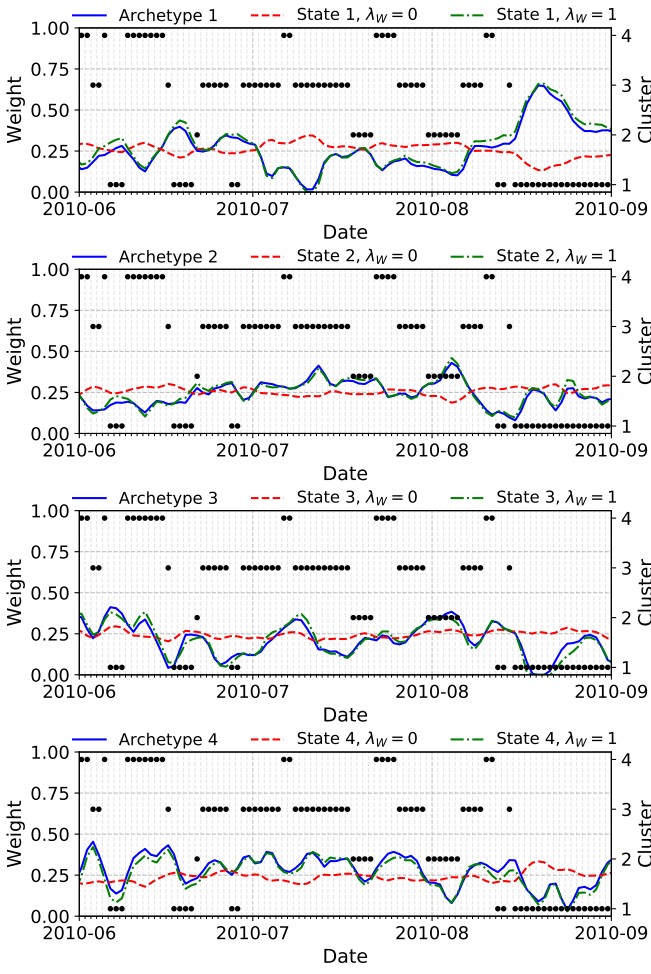

**Figure 14.** Time-series of basis weights (lines) associated with each archetype produced by AA and the states produced by convex coding with and without regularization, for $k = 4$ states during the 2010 boreal summer. The corresponding $k$-means cluster assignments for each day are shown as points. Note that the monthly mean $Z'_{g500\mathrm{hPa}}$ for July 2010 associated with the heat wave event most closely resembles cluster 3 in the $k$-means clustering, to which most of the daily anomalies for that month are also assigned.





In contrast, despite the severity of this event, individual daily height anomalies during July 2010 are not unambiguously iden-
tified as extremes by AA or the less constrained convex coding. In Fig. 14 we show the time-series of weights associated with

the basis vectors found by AA and by convex coding either without regularization, $\lambda_W = 0$, or with regularization parameter
$\lambda_W = 1$, which for these data yields dictionary vectors very similar to those found by AA. For all three cases, roughly equal
weights are assigned to each basis vector during July 2010. The complete anomaly is therefore represented as a mixture of all of
the dictionary vectors, rather than being assigned to a single characteristic type of extreme. Evidently, any single basis pattern
extracted by these methods does not directly correspond to a particular indicator of extreme weather conditions; extreme events

of practical concern may be spread across multiple basis vectors, making identifying such events in this representation more
difficult. The hard clustering obtained using $k$-means is perhaps more easily interpreted in this case, as the resulting cluster
affiliations show frequent occurrences of the European blocking cluster (i.e., cluster 3 in Fig. 12 and Fig. 14) during the peak
heat-wave period, with a smaller number of days assigned to clusters 2 and 4, suggesting a clearer picture of residence in a
single, persistent blocking state. On the other hand, the coarse representation of the actual anomalies by the single cluster 3

centroid is poor compared to the reconstruction provided by AA and convex coding, but could be improved by, for example,
making use of a soft clustering algorithm instead. As in the SST case study, the ambiguous classification of events provided by
the convex hull based methods is less of a problem when state space location alone (e.g., temperature anomalies) is in itself a
relevant feature. When this is not the case, as here, methods that take advantage of state space density, either due to recurrence
or persistence, may be more easily interpreted, or alternatively hybrid approaches could be used in order to partition the state

space.

## 4   Conclusions

Representing a high-dimensional dataset in terms of a highly reduced basis or dictionary is an essential step in many climate
analyses. Beyond the practical necessity of doing so, it is usually also desirable for the individual elements of the representation
to be identifiable with physically relevant features for the sake of interpretation. A wide range of popular dimension reduction

methods, including PCA, $k$-means clustering, AA, and convex coding, can be written down as particular forms of a basic matrix
factorization problem. These methods differ in the details of the measure of cost that is optimized and the feasible solution
regions, with the result that different methods yield representations suitable for targeting different features of the data. Of the
methods considered here, PCA extracts directions in state space corresponding to maximal variability, $k$-means locates central
points, and AA and similar convex codings identify points on or outside the convex hull of the observed data and so find a

representation in terms of extreme points in state space. As different features may be relevant for different applications, it is
important to consider these distinctions between these factorization methods and carefully choose a method that is effective for
extracting the features of interest.

In some cases, the representations obtained using different dimension reduction methods are very similar, and one can
identify more or less easily interpretable features using any given method. This is exemplified by our first case study of SST

anomalies, in which the presence of the dominant ENSO mode ensures that PCA, $k$-means clustering, AA, and convex coding



all identify similar bases corresponding to well-known physical modes. In this case, the main distinction between the methods arises in the nature of the classification of the data (e.g., hard versus soft clustering) and the accuracy with which the original data can be reconstructed for a given level of compression.

As our second case study demonstrates, neglecting the important role played by temporal persistence in dynamically relevant features can lead to representations that are difficult to interpret and may not be as effective for studying persistent states. Clustering-based approaches, or more generally methods that attempt to approximate modes in the PDF rather than targeting the tails of the distribution, are likely to be a better choice in these circumstances. This can also be achieved by appropriate regularization so as to reduce sensitivity to transient features or outliers, which otherwise drive the definition of the basis in methods such as AA. In all of the methods that we have considered, a lack of independence in time is not explicitly modeled. Extensions to the simple methods that we have considered to account for non-independence are also possible, albeit usually at the cost of increased complexity. Singular spectrum analysis (see, e.g., Jolliffe (1986)) is a familiar example of one such extension for PCA. Similar generalizations can also be constructed for convex decompositions of the data. For instance, by virtue of the decomposability of the least squares cost function, it is possible to construct a joint convex discretization of instantaneous and lagged values of the variables of interest, which forms the basis of the scalable probabilistic approximation method (Gerber et al., 2020). In this approach, the decomposition may be parametrized in terms of a transition matrix relating the weights at different times, thus naturally incorporating a temporal constraint into the discretization. An associated regularization parameter allows the amount of temporal regularization to be appropriately tuned. Moreover, because the optimization problem remains separable in this case, the individual optimization steps can be parallelized and the method remains scalable. More sophisticated regularization strategies (Horenko, 2020) can further improve upon the performance of the method, even in the case of relatively small sample sizes and feature degeneracies.

The idea of imposing temporal regularization via assumed dynamics for the latent weights suggests that another approach to better target particular features is to start from an appropriately defined generative model. An underlying probabilistic model is already suggested by the stochastic constraints that are imposed on the weights in AA and in convex coding, a feature that is already taken advantage of in the case of the scalable probabilistic approximation. Corresponding latent variable models can naturally be constructed for PCA (Roweis, 1998; Tipping and Bishop, 1999) and AA (Seth and Eugster, 2016). From the point of view of incorporating temporal dependence between samples, starting from a probabilistic model is fruitful as it is conceptually straightforward to incorporate a model for the dynamics of the latent weights $z_t$ (Lawrence, 2005; Wang et al., 2006; Damianou et al., 2011). Taken together with a choice of conditional distribution for the observations, point estimation in the latent variable model is once again achieved by optimization of a suitable loss function. Regularization is in this context provided by suitable choice of prior distributions for the dictionary and weights. An additional advantage of starting from a generative model is the possibility of applying the full machinery of Bayesian inference rather than obtaining only point estimates (Mnih and Salakhutdinov, 2008; Salakhutdinov and Mnih, 2008; Virtanen et al., 2008; Shan and Banerjee, 2010; Gönen et al., 2013), although scaling such analyses remains a challenging issue. While this approach appears to be natural for constructing dimension reduction methods that may flexibly take into account more complicated dependence structures, it is by no means guaranteed to provide improved low-dimensional representations of the data, and likely depends heavily on choices



such as the assumed generative process for the weights. For example, in simple constructions with weights drawn according to a Gaussian process, in the absence of strong prior information the fitted bases are driven to be very similar to those found by ordinary PCA in order to maximize the likelihood of the observed data, while at the same time being substantially more expensive to fit. Thus, the development of temporally regularized methods derived from underlying stochastic models remains

a topic for further investigation.

*Code and data availability.* The HadISST SST dataset used in this study is provided by the UK Met Office Hadley Centre and may be accessed at https://www.metoffice.gov.uk/hadobs/hadisst/. The JRA-55 geopotential height data used is made available through the JRA-55 project and may be accessed following the procedures described at https://jra.kishou.go.jp/JRA-55/index_en.html. All source code used to perform the analyses presented in the main text may be found at https://doi.org/10.5281/zenodo.3723948.

*Author contributions.* All of the authors designed the study. TO proposed the specific case studies, and DH implemented the code to perform the numerical comparisons and generated all plots and figures. All of the authors contributed to the direction of the study, discussion of results, and the writing and approval of the manuscript.

*Competing interests.* The authors declare that they have no conflicts of interest.

*Acknowledgements.* The authors would like to thank Illia Horenko for continual guidance and for his valuable inputs over the course of this
project, and Vassili Kitsios and Didier Monselesan for encouragement and helpful discussions about the various methods. DH is supported by the Australian Commonwealth Scientific and Industrial Research Organisation (CSIRO) through a ResearchPlus postdoctoral fellowship. TO is supported by the CSIRO Decadal Climate Forecasting Project (https://research.csiro.au/dfp). The PCA and $k$-means results, and the visualizations using metric MDS, were obtained using the routines provided by the scikit-learn Python package (Pedregosa et al., 2011). Plots were generated using the Python package Matplotlib (Hunter, 2007).





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
