# Peer review of "Applications of matrix factorization methods to climate data"

_Nonlinear Processes in Geophysics, 2020_

## Referee Comment (RC1) · Anonymous Referee #1 · 12 Jun 2020

This paper is a cross between a review and evaluation: it compares the theoretic underpinnings of several established methods for dimension reduction and offers example applications to illustrate the differences. Though the techniques are not new, the work does offer interesting perspectives, and the paper is mostly well-written and well illustrated. A few suggestions are summarized here:

1) it is helpful to use a table to succinctly summarize the key differences among the techniques. Some of the algorithmic details are unnecessarily elaborated (e.g., pg 8) whereas the actual differences are obscured. For example, does K-mean cluster produce orthogonal basis vectors? and are the clusters easier to interpret than principal components of PCA. What are the unique advantages of AA relative to PCA and K-mean cluster?

[Figure]

2) please explain how the case studies were chosen. Are the outcomes of the case studies supposed to inform us about the geophysical variables that one, or several of the approaches are more suitable than others?

3) For the SST case, I wonder what the take home message is in terms of the difference among the three methods as illustrated in Figs. 3-9? Fig 10 shows that the PCA features lower RMSE than others and yet the conclusion appears to be that these methods are all comparable.

4) For the Z500 anomaly case, what is the recommendation of lambda_w? And what methods offer a clear linkage between the resulting patterns and "physical extremes"?

---

## Referee Comment (RC2) · Anonymous Referee #2 · 20 Jul 2020

This study compares several dimension reduction methods in scenarios with or without a dominant large-scale mode separating from the smaller scales. The introduction section provides a nice review of different dimension reductions methods for finding dominant modes of climate variability. The paper is mostly well written and results are interesting and provides useful suggestions for choosing a proper method for different climate variability analysis scenarios. I have only a few minor comments listed as follows.

1. The conclusion section does mention the caveat of neglecting the role of time dimension. The SST case has a dominant ENSO signal that is mostly a time oscillation of a fixed spatial pattern, while in the geopotential height case there are traveling wave signals with spatially changing patterns. For the later case, can the time dimension be

included in the analysis so that a more physically interpretable mode can be found? Can including time dimension directly in the d-dimensional data produce different results from applying temporal regularization?

2. From the description in section 2, it is a little hard to conceptualize the key differences between the AA and CC methods, and their advantages over the k-mean clustering method. Could you list the cost function and constraints for each method in a succinct manner and highlight their differences?

3. Comparing Figures 9 and 13, the behavior of CC and AA finding basis with more extreme departures from mean than k-mean clustering is the same for both the SST and Z cases. Does one case prefer bases with larger departures from mean than the other? Or, is the magnitude of bases functions less important than their alignment with the actual physical modes?

4. For the Z case, both AA and CC methods are not aligned with the PCA bases functions. Do you have any insights of which method is superior in this case? Or, are they all not finding the physical modes because of missing the time dimension?

5. For the RMSE for reconstructed data (Fig 10), is there a similar plot for the geopotential height case?

---

## Author Comment (AC1) · 23 Jul 2020

Dear Reviewer,

Thank you very much for your positive comments and suggestions. The suggestions provided we agree improve the presentation of several of the key points in the study, and we have attempted to address each in turn. Please find below our responses to the individual comments and a description of the corresponding changes to the manuscript made to address each point. For each point, the initial comment is given in bold, while modifications to the text are italicised and shown in blue.

1. **it is helpful to use a table to succinctly summarize the key differences among the techniques. Some of the algorithmic details are unnecessarily elaborated (e.g., pg8) whereas the actual differences are obscured. For example, does K-mean cluster produce orthogonal basis vectors? and are the clusters easier to interpret than principal components of PCA. What are the unique advantages of AA relative to PCA and K-mean cluster?**

   We agree that it is helpful to have a single summary of the key differences between the methods, and that too much focus was placed on the numerical solution of the optimization problem. To address this, the discussion of the numerical implementation of the solution for the AA/CC optimization problem beginning on line 191 of the original manuscript has been moved to a separate appendix. Instead, we now conclude Sect. 2 with a summary of the key differences between the different methods, including a table (Table 1) collecting the different cost and constraint functions, that reads as follows:

   *The various choices of cost function and constraints defining the above methods are summarized in Table 1. While all of the methods fit within the broader class of matrix factorizations, the different choices of cost func-*

Table 1: Summary of the definitions of each of the four methods compared in this study. Each method is defined by a choice of cost function to be minimized together with a set of constraints placed on the factors $Z$ and $W$ (or $Z$ and $C$ for AA). The choices for each impact that nature of the features that each method extracts from a particular dataset.

| Method | Cost function | Constraints | Targeted features |
|---|---|---|---|
| PCA | $\frac{1}{2T}\|X - ZW^T\|_F^2$ | $\mathrm{rank}(ZW^T) \leq k$ | Directions of maximum variance |
| $k$-means | $\frac{1}{2T}\|X - ZW^T\|_F^2$ | $Z_{ti} \in \{0,1\}$, $Z\mathbf{1}_k = \mathbf{1}_T$ | Data centroids |
| Convex coding | $\frac{1}{2T}\|X - ZW^T\|_F^2 + \lambda_W \Phi_W(W)$ | $Z \succeq 0$, $Z\mathbf{1}_k = \mathbf{1}_T$ | Basis convex hull |
| AA | $\frac{1}{2T}\|X - ZCX\|_F^2$ | $C \succeq 0$, $C\mathbf{1}_T = \mathbf{1}_k$, $Z \succeq 0$, $Z\mathbf{1}_k = \mathbf{1}_T$ | Data convex hull |

   *tions and constraints lead to important differences in the low-dimensional representation of the data produced by each method. For instance, in contrast to PCA, the basis vectors produced by k-means, convex coding, or AA are in general not orthogonal. In some circumstances, this non-orthogonality may be advantageous when the structure necessary to ensure orthogonal basis vectors (e.g., via appropriate cancellations) obscures important features or makes interpretation of the full PCA basis vectors difficult. A k-means clustering may, for example, provide a much more natural reduction of the data when multiple distinct, well-defined clusters are present. The cost function and choice of constraints that define convex coding and archetypal analysis imply that the optimal basis vectors produced by these methods are such that their convex hull (i.e., the set of all linear combinations of the basis vectors with weights summing to one) best fits the data. Consequently, both are well-suited for describing data where points can be usefully characterized in terms of their relationship to a set of extreme values, be they spatial patterns of large positive or negative anomalies in a geophysical field or particular combinations of spectral components in a frequency domain representation of a signal. PCA and k-means may be less useful in such cases, as neither yields a decomposition of the data in terms of points at or outside of the boundaries of the observations. AA differs from more general convex encodings in imposing the stricter requirement that the dictionary elements, i.e., the archetypes, lie on the boundary of the data. It is, in this sense, conservative, in that the features extracted by AA lie on the convex hull of the data and so correspond to a set of extremes that are nevertheless consistent with the observed data. In the absence of any regularization ($\lambda_W \to 0$), the general convex codings that we consider admit basis vectors that lie well outside of the observed data. By doing so, the method finds a set of basis vectors whose convex hull better reconstructs the data than in AA, at the cost of representing it in terms of points that may not be physically realistic. This behaviour, and the impact of the different choices of cost function and constraints, is sketched in Fig. 1.*

2. **please explain how the case studies were chosen. Are the outcomes of the case studies supposed to inform us about the geophysical variables that one, or several of the approaches are more suitable than others?**

Yes, this is correct; the two case studies were chosen to highlight instances where some of the methods may not be as suitable or useful as the others, depending on the focus of the analysis. In the case of the SST data, the existence of a single, well-separated mode of variability results in all four methods producing similar bases with which to represent the data. Consequently, for this application all of the methods are relatively comparable when it comes to defining a set of modes, and are distinguished by other features such as the magnitude of the reconstruction error for a given basis size, as noted in point 3. The second case study is chosen to emphasize the fact that this agreement between the methods is, however, dependent on the features of the SST data, so that for variables where this scale separation is absent, such as geopotential height, the choice of method becomes more important from the point of view of extracting useful modes. To attempt to summarize this motivation for the choice of case studies, the first paragraph of Sect. 3 has been expanded and now reads as follows:

*We now turn to a set of case-studies that demonstrate some of the implications of the various differences noted above in realistic applications. We consider two particular examples that highlight the importance of considering the particular physical features of interest when choosing among possible dimension reduction methods. The first example that we consider, an analysis of SST anomalies, is characterized by a large separation of scales between modes of variability together with key physical modes, particularly ENSO, that can be directly related to extreme values of SST anomalies. This means that the basis vectors or spatial patterns extracted by PCA, k-means, and convex coding are for the most part rather similar in structure, and so the choice of method may be guided by other considerations, such as the level of reconstruction error. We then contrast this scenario with the example of an analysis of mid-latitude geopotential height anomalies, where there is neither scale separation nor do the physical modes coincide solely with extreme anomaly values. As a result, methods that are based on constructing a convex coding of the data, without targeting features that capture the dynamical characteristics of the relevant modes, produce representations that are, arguably, more difficult to interpret, and hence may be less suitable than clustering based methods.*

3. **For the SST case, I wonder what the take home message is in terms of the difference among the three methods as illustrated in Figs. 3-9? Fig 10 shows that the PCA features lower RMSE than others and yet the conclusion appears to be that these methods are all comparable.**

We agree that the take home point was not clearly made in the SST case. For the SST data, all four methods are comparable in the sense that the physical space patterns that are identified as basis vectors are similar, as illustrated in Figs. 3 - 9, and so no particular method is obviously preferable for identifying relevant modes. Methods that are based on locating the convex hull of the data in this case perhaps provide a more direct interpretation of extreme events, which can be characterized for SSTs in terms of the magnitudes of the anomalies, but the patterns produced do not differ dramatically from ordinary PCA. As rightly noted, though, the latter also provides the optimal RMSE for a given basis size, and so is distinguished from the other methods in terms of the fidelity of the reconstruction; indeed, this behavior in terms of reconstruction error is generic and we should have highlighted this detail in the first version of the article. To address this point, we now summarize our conclusions from the case study at the end of Sect. 3.1, beginning at line 311 of the original manuscript, as follows:

*The ordering of the methods in terms of reconstruction error observed in Fig. 10 is expected to be the case more generally. As noted in Sect. 2, PCA provides the globally optimal reconstruction of the data matrix with a given rank, in the absence of any constraints, and so amounts to a lower bound on the achievable reconstruction error. Of the remaining methods, the additional freedom to locate the basis elements outside of the convex hull of the data when performing an unregularized convex coding allows for a lower reconstruction error than is achievable using archetypes. For a given basis size the hard clustering resulting from k-means generally results in the largest RMSE. For larger values of the regularization $\lambda_W$, the optimal basis elements sit within the convex hull of the data and provide a fuzzy representation of the data with a progressively increasing RMSE. In this particular analysis of SST data, the performance of the different methods with respect to reconstruction error is one distinguishing factor that may guide the choice of method; while all four produce similar large-scale spatial patterns, for a given basis size PCA provides the lowest reconstruction error and might be preferred if information loss is a significant concern.*

4. **For the Z500 anomaly case, what is the recommendation of lambda_w? And what methods offer a clear linkage between the resulting patterns and "physical extremes"?**

In general, the choice of the regularization parameter will depend on the particular use case, a point which we agree should have been made more clearly. For our purposes, the values $\lambda_W = 0$ and $\lambda_W = 10$ were chosen to provide an illustration of the impact of this parameter on the fitted bases, and in particular to highlight the role of the regularization in producing bases that vary from minimizing the reconstruction error to focusing on features in the data that are less impacted by noise. In the geopotential height case, we argue that placing more emphasis on the latter provides clearer linkage to the physical extremes, as doing so tends to prefer states that exhibit some degree of persistence and thus matches the characteristics of the physical features. This is also achieved by the use of $k$-means, whereas AA in its usual formulation takes into account only those points lying on the convex hull of the data and so does not provide a particularly good characterization of the physical extremes in terms of the associated patterns. It should be noted of course that the dimension reductions considered here are a starting point for further analysis, e.g., causally relating the observed weather patterns to known extreme events requires further dynamical analysis, but methods that identify patterns that closely resemble the relevant structures provide clearer starting points for such analyses. To address this, we have expanded the discussion (starting at line 386 of the original manuscript) at the end of Sect. 3.2 to summarize these observations:

*. . . but could be improved by, for example, making use of a soft clustering algorithm instead. To summarize, in the geopotential height case where extreme events are defined not just by large anomalies but persistent structures, methods such as k-means, or the more heavily regularized convex coding applied here, that better identify such structures provide bases that are more amenable to interpretation in terms of physical extremes and so may provide a more direct starting point for analyses of such events. Unregularized convex coding and archetypal analysis, on the other hand, are less well suited in this respect, as they do not yield a direct assignment of such events to individual states.*

*Finally, it is worth noting that, as in the SST case study, . . .*

In practice, the regularization $\lambda_W$ can be chosen using standard model selection criteria depending on the user's objective; for instance, cross-validation could be used to select a value of $\lambda_W$ that provides the best out-of-sample reconstruction or prediction error. As this was not pointed out in the original manuscript, we have now added the following comment at line 359 to address the problem of choosing $\lambda_W$:

*In this sense, by appropriate choice of regularization it is possible to interpolate between a representation of the data in terms of points on the convex hull and mean features, depending on how much weight is placed on minimizing the reconstruction error. While here we consider only a few levels of regularization for illustrative purposes, in general the degree to which this is done, i.e., the choice of $\lambda_W$, can be guided by standard model selection methods, such as cross-validation.*

Additionally, we have made the following minor changes:

- On line 6 of the original manuscript, in the abstract the use of the acronym "EOF" has been replaced by "empirical orthogonal function".

- Starting on line 21 of the original manuscript, the description of empirical orthogonal functions has been expanded to read: *Perhaps the most familiar example in climate science is provided by empirical orthogonal function (EOF; Lorenz (1956); Hannachi et al. (2007)) or principal component analysis (PCA; Jolliffe (1986)), which identifies directions of maximum variance in the data, or, more generally, the directions maximizing a chosen norm.*

- For clarity, on line 314 it is highlighted that the bases produced by the methods correspond to spatial patterns, and now reads: *As a result, all of the dimension reduction methods that we consider extract similar bases (patterns) . . .*

---

## Author Comment (AC2) · 23 Jul 2020

**Dear Reviewer,**

Thank you very much for your positive and constructive comments. The points raised focus on several key aspects of the study, and thus we believe that in addressing them the clarity and usefulness of the article has been greatly improved. Please find below our responses to the individual comments and a description of the corresponding changes to the manuscript made to address each point. For each point, the initial comment is given in bold, while modifications to the text are italicised and shown in blue.

1. The conclusion section does mention the caveat of neglecting the role of time dimension. The SST case has a dominant ENSO signal that is mostly a time oscillation of a fixed spatial pattern, while in the geopotential height case there are traveling wave signals with spatially changing patterns. For the later case, can the time dimension be included in the analysis so that a more physically interpretable mode can be found? Can including time dimension directly in the d-dimensional data produce different results from applying temporal regularization?

In general, explicitly including information about time-dependence in the dimension reduction method may produce better results in terms of extracting more physically interpretable modes, but in practice whether or not this is actually achieved will depend on the details of how the time dimension is incorporated. It is also correct that this may produce different results from applying some forms of temporal regularization, although as regularizing terms can be freely constructed the comparison between the two approaches has to be done on a case-by-case basis. Extending the definitions of the various methods to explicitly model time-dependence is an option that we had intended to include in the paragraph beginning on line 425 of the original manuscript, but we believe that the description given there is too narrow. To address this, we have extended the opening sentence to clarify that we include models that include some sort of explicit time-dependence, which now reads

The idea of imposing temporal regularization via assumed dynamics for the latent weights suggests that another approach to better target particular features is to start from an appropriately defined generative model, or otherwise explicitly incorporating appropriate time-dependence when constructing a reduction method.

**2. From the description in section 2, it is a little hard to conceptualize the key differences between the AA and CC methods, and their advantages over the k-mean clustering method. Could you list the cost function and constraints for each method in a succinct manner and highlight their differences?**

We agree that the description of the various methods given in Sect. 2 does not provide a sufficiently clear summary of the key distinctions between the possible choices, and puts too much emphasis on numerical details. To address this, we have moved the discussion of the numerical implementation beginning on line 191 to a separate appendix, and instead now conclude Sect. 2 with the following brief summary of the different methods that highlights the relevant choices that enter into the definition of each:

The various choices of cost function and constraints defining the above methods are summarized in Table 1. While all of the methods fit within the broader class of matrix factorizations, the different choices of cost func-

Table 1: Summary of the definitions of each of the four methods compared in this study. Each method is defined by a choice of cost function to be minimized together with a set of constraints placed on the factors Z and W (or Zand C for AA). The choices for each impact that nature of the features that each method extracts from a particular dataset.

| uataset.      |                                                       |                                                    |                                |
|---------------|-------------------------------------------------------|----------------------------------------------------|--------------------------------|
| Method        | Cost function                                         | Constraints                                        | Targeted features              |
| PCA           | $\frac{1}{2T} \ X - ZW^T\ _F^2$                       | $\operatorname{rank}(ZW^T) \le k$                  | Directions of maximum variance |
| k-means       | $\frac{1}{2T} \ X - ZW^T\ _F^2$                       | $Z_{ti} \in \{0,1\},  Z1_k = 1_T$                  | Data centroids                 |
| Convex coding | $\frac{1}{2T} \ X - ZW^T\ _F^2 + \lambda_W \Phi_W(W)$ | $Z \succeq 0, Z 1_k = 1_T$                         | Basis convex hull              |
| AA            | $\frac{1}{2T} \ X - ZCX\ _F^2$                        | $C \succeq 0, C1_T = 1_k, Z \succeq 0, Z1_k = 1_T$ | Data convex hull               |

tions and constraints lead to important differences in the low-dimensional representation of the data produced by each method. For instance, in contrast to PCA, the basis vectors produced by k-means, convex coding, or AA are in general not orthogonal. In some circumstances, this non-orthogonality may be advantageous when the structure necessary to ensure orthogonal basis vectors (e.g., via appropriate cancellations) obscures important features or makes interpretation of the full PCA basis vectors difficult. A k-means clustering may, for example, provide a much more natural reduction of the data when multiple distinct, well-defined clusters are present. The cost function and choice of constraints that define convex coding and archetypal analysis imply that the optimal basis vectors produced by these methods are such that their convex hull (i.e., the set of all linear combinations of the basis vectors with weights summing to one) best fits the data. Consequently, both are well-suited for describing data where points can be usefully characterized in terms of their relationship to a set of extreme values, be they spatial patterns of large positive or negative anomalies in a geophysical field or particular combinations of spectral components in a frequency domain representation of a signal. PCA and k-means may be less useful in such cases, as neither yields a decomposition of the data in terms of points at or outside of the boundaries of the observations. AA differs from more general convex encodings in imposing the stricter requirement that the dictionary elements, i.e., the archetypes, lie on the boundary of the data. It is, in this sense, conservative, in that the features extracted by AA lie on the convex hull of the data and so correspond to a set of extremes that are nevertheless consistent with the observed data. In the absence of any regularization ( $\lambda_W \to 0$ ), the general convex codings that we consider admit basis vectors that lie well outside of the observed data. By doing so, the method finds a set of basis vectors whose convex hull better reconstructs the data than in AA, at the cost of representing it in terms of points that may not be physically realistic. This behaviour, and the impact of the different choices of cost function and constraints, is sketched in Fig. 1.

**3. Comparing Figures 9 and 13, the behavior of CC and AA finding basis with more extreme departures from mean than k-mean clustering is the same for both the SST and Z cases. Does one case prefer bases with larger departures from mean than the other? Or, is the magnitude of bases functions less important than their alignment with the actual physical modes?**

We tend to agree that, as far as the discussion in the article goes, the magnitude of the basis functions is less important than the alignment of the various basis choices, although it is certainly not immaterial. There are two points that we should have explained more clearly in the original manuscript. Firstly, with respect to the common behavior of AA and CC choosing a dictionary representing larger departures from the mean state than k-means, this is expected from the definitions of the methods: AA and CC are designed to find points that lie at the "boundary" of the observed data, in the case of AA, or outside of it when using a general convex coding. As a result, both methods will select bases further from the mean than k-means, which selects the cluster means as a dictionary. The extent to which CC chooses more extreme basis points than AA will depend on the particular characteristics of the dataset, and so it is difficult to compare the extent to which one case may show larger departures from the mean than another. However, it is right to note that such large departures are obtained with CC, and so to try better emphasize this point we have included this observation in the summary paragraph concluding Sect. 2, where it is pointed out that the resulting basis vectors may not be physically realistic. For the purposes of the case studies, we would argue that the main distinction is the fact that, for the geopotential height case, the bases do not correspond to the PCA basis (or the physical modes) and so are more difficult to make use of. To try to clarify this point, we have expanded the third paragraph of Sect. 3.2 to highlight some of the similarities and differences. Beginning on line 339 of the original manuscript, these changes read:

The relative dispositions of each of the states with respect to each other, as measured by Euclidean distance, are visualized in Fig. 13 on the basis of a two-dimensional metric MDS. In some respects, the performance of the different methods is similar to that seen in the SST example; for example, as expected on general grounds the ordering of the methods with respect to achieved RMSE (not shown) is the same as in Fig. 10. Similarly, AA and the unregularized convex coding select, by design, basis vectors that lie on or outside of the convex hull of the data, with the latter having the freedom to choose a basis corresponding to much larger departures from the mean so as to reduce the reconstruction error; note that the precise degree to which the basis vectors lie outside the convex hull will depend on the particular data at hand, but the behavior is otherwise generic. However, unlike the SST case, ...

4. For the Z case, both AA and CC methods are not aligned with the PCA bases functions. Do you have any insights of which method is superior in this case? Or, are they all not finding the physical modes because of missing the time dimension?

Yes, neither AA nor CC are finding the expected modes as they do not account for the important role of persistence in defining these modes. We agree that the discussion of which method is superior, or at least more useful, in the case of the geopotential height anomalies was not clearly done. In this case, we argue that the regularized convex coding is more appropriate for this case, as it better targets the persistent features characterizing atmospheric extremes, along with k-means. While they also do not explicitly include the timedimension, they do pick up persistent or quasi-stationary features, and hence give a better extraction of the large-scale anomaly structures. To try to clarify this point, we have added a short summary at the end of Sect. 3.2, beginning at line 386 of the original manuscript, to read:

... but could be improved by, for example, making use of a soft clustering algorithm instead. To summarize, in the geopotential height case where extreme events are defined not just by large anomalies but persistent structures, methods such as k-means, or the more heavily regularized convex coding applied here, that better identify such structures provide bases that are more amenable to interpretation in terms of physical extremes and so may provide a more direct starting point for analyses of such events. Unregularized convex coding and archetypal analysis, on the other hand, are less well suited in this respect, as they do not yield a direct assignment of such events to individual states.

Finally, it is worth noting that, as in the SST case study, ...

**5. For the RMSE for reconstructed data (Fig 10), is there a similar plot for the geopotential height case?**

The equivalent plot for the geopotential height case study is shown in Figure 1. The ordering of the methods

Figure 1: RMSE for reconstructing the leading 167 PCs of geopotential height anomalies resulting from each of the methods.

with respect to the reconstruction RMSE is the same as in the SST case shown in Figure 10, although note that since the clustering in this case is performed on the PCs themselves for efficiency there is not an equivalent line for PCA. This ordering of the methods in terms of reconstruction error is expected on general grounds as briefly discussed following Figure 10 (lines 304 - 311); that is, for a given dataset, PCA yields the globally minimal reconstruction error as guaranteed by the Eckhart-Young theorem, while the unconstrained convex coding can approach the same level of fidelity by choosing basis points sufficiently far outside the convex hull of the data. AA and k-means, being more constrained, in turn have a larger reconstruction error for a given basis size, with AA typically performing better than k-means in this respect by virtue of being able to provide a soft-clustering of the data. As this behavior is the same in both the SST and geopotential height case, we do not include Figure 1 in the article to reduce repetition. However, we do think that we should have better emphasized the expected performance of the different methods in terms of reconstruction error, as this is an important point to consider when, e.g., one is interested in simply achieving a compression of the data with minimal loss of information. To address this point, we have firstly expanded the discussion in the last paragraph of Sect. 3.1 to better explain this point, with the following additions beginning on line 311:

The ordering of the methods in terms of reconstruction error observed in Fig. 10 is expected to be the case more generally. As noted in Sect. 2, PCA provides the globally optimal reconstruction of the data matrix with a given

rank, in the absence of any constraints, and so amounts to a lower bound on the achievable reconstruction error. Of the remaining methods, the additional freedom to locate the basis elements outside of the convex hull of the data when performing an unregularized convex coding allows for a lower reconstruction error than is achievable using archetypes. For a given basis size the hard clustering resulting from k-means generally results in the largest RMSE. For larger values of the regularization  $\lambda_W$ , the optimal basis elements sit within the convex hull of the data and provide a fuzzy representation of the data with a progressively increasing RMSE. In this particular analysis of SST data, the performance of the different methods with respect to reconstruction error is one distinguishing factor that may guide the choice of method; while all four produce similar large-scale spatial patterns, for a given basis size PCA provides the lowest reconstruction error and might be preferred if information loss is a significant concern.

We also now explicitly note in the text that the same ordering in terms of RMSE is observed in the geopotential height case; this is included in the additions in response to point 3 above.

Additionally, we have also made the following minor changes to the manuscript:

- On line 6 of the original manuscript, in the abstract the use of the acronym "EOF" has been replaced by "empirical orthogonal function".
- Starting on line 21 of the original manuscript, the description of empirical orthogonal functions has been expanded to read: Perhaps the most familiar example in climate science is provided by empirical orthogonal function (EOF; Lorenz (1956); Hannachi et al. (2007)) or principal component analysis (PCA; Jolliffe (1986)), which identifies directions of maximum variance in the data, or, more generally, the directions maximizing a chosen norm.
- For clarity, on line 314 it is highlighted that the bases produced by the methods correspond to spatial patterns, and now reads: As a result, all of the dimension reduction methods that we consider extract similar bases (patterns) ...